# Phytochemical Profiling of Extracts from Rare *Potentilla* Species and Evaluation of Their Anticancer Potential

**DOI:** 10.3390/ijms24054836

**Published:** 2023-03-02

**Authors:** Daniel Augustynowicz, Marta Kinga Lemieszek, Jakub Władysław Strawa, Adrian Wiater, Michał Tomczyk

**Affiliations:** 1Department of Pharmacognosy, Faculty of Pharmacy with the Division of Laboratory Medicine, Medical University of Białystok, ul. Mickiewicza 2a, 15-230 Białystok, Poland; 2Department of Medical Biology, Institute of Rural Health, ul. Jaczewskiego 2, 20-090 Lublin, Poland; 3Department of Industrial and Environmental Microbiology, Institute of Biological Sciences, Maria Curie-Skłodowska University, ul. Akademicka 19, 20-033 Lublin, Poland

**Keywords:** *Potentilla*, Rosaceae, polyphenols, LC–HRMS, colorectal cancer, LS180 cells, cytotoxicity, CCD841 CoN cells

## Abstract

Despite the common use of *Potentilla* L. species (Rosaceae) as herbal medicines, a number of species still remain unexplored. Thus, the present study is a continuation of a study evaluating the phytochemical and biological profiles of aqueous acetone extracts from selected *Potentilla* species. Altogether, 10 aqueous acetone extracts were obtained from the aerial parts of *P. aurea* (**PAU7**), *P. erecta* (**PER7**), *P. hyparctica* (**PHY7**), *P. megalantha* (**PME7**), *P. nepalensis* (**PNE7**), *P. pensylvanica* (**PPE7**), *P. pulcherrima* (**PPU7**), *P. rigoi* (**PRI7**), and *P. thuringiaca* (**PTH7**), leaves of *P. fruticosa* (**PFR7**), as well as from the underground parts of *P. alba* (**PAL7r**) and *P. erecta* (**PER7r**). The phytochemical evaluation consisted of selected colourimetric methods, including total phenolic (TPC), tannin (TTC), proanthocyanidin (TPrC), phenolic acid (TPAC), and flavonoid (TFC) contents, as well as determination of the qualitative secondary metabolite composition by the employment of LC–HRMS (liquid chromatography–high-resolution mass spectrometry) analysis. The biological assessment included an evaluation of the cytotoxicity and antiproliferative properties of the extracts against human colon epithelial cell line CCD841 CoN and human colon adenocarcinoma cell line LS180. The highest TPC, TTC, and TPAC were found in **PER7r** (326.28 and 269.79 mg gallic acid equivalents (GAE)/g extract and 263.54 mg caffeic acid equivalents (CAE)/g extract, respectively). The highest TPrC was found in **PAL7r** (72.63 mg catechin equivalents (CE)/g extract), and the highest TFC was found in **PHY7** (113.29 mg rutin equivalents (RE)/g extract). The LC–HRMS analysis showed the presence of a total of 198 compounds, including agrimoniin, pedunculagin, astragalin, ellagic acid, and tiliroside. An examination of the anticancer properties revealed the highest decrease in colon cancer cell viability in response to **PAL7r** (IC_50_ = 82 µg/mL), while the strongest antiproliferative effect was observed in LS180 treated with **PFR7** (IC_50_ = 50 µg/mL) and **PAL7r** (IC_50_ = 52 µg/mL). An LDH (lactate dehydrogenase) assay revealed that most of the extracts were not cytotoxic against colon epithelial cells. At the same time, the tested extracts for the whole range of concentrations damaged the membranes of colon cancer cells. The highest cytotoxicity was observed for **PAL7r**, which in concentrations from 25 to 250 µg/mL increased LDH levels by 145.7% and 479.0%, respectively. The previously and currently obtained results indicated that some aqueous acetone extracts from *Potentilla* species have anticancer potential and thus encourage further studies in order to develop a new efficient and safe therapeutic strategy for people who have been threatened by or suffered from colon cancer.

## 1. Introduction

Cancer, a non-infectious disease, is one of the most dreadful diagnoses that severely impacts a patient’s life quality. Unfortunately, cancer is a significant and increasing cause of death worldwide. The European Cancer Information System (ECIS) estimated an increase in new cases of cancer in the European Union (EU-27) from 2.68 million in 2020 to 3.24 million in 2040, a 21% increase, while the cancer-related death toll is estimated to increase from 1.26 million to 1.66 million cases, a 31.8% increase. Colorectum cancer is the second-most-diagnosed cancer type in EU-27 countries, with over 0.34 million cases in 2020; however, in 2040, it will overtake breast cancer as the most commonly diagnosed cancer type with over 0.43 million cases [1]. The most frequently used method to treat early-stage colorectal cancer is surgical resection, which effectively relieves the patient’s symptoms. However, approximately 25% to 30% of patients after successful surgery will develop metastases within 5 years [2]. Moreover, in the further stages, unresectable metastatic cancer systemic therapy includes chemotherapy, radiotherapy, immunotherapy, and biological therapy, such as antibodies to cellular growth factors, as well as their combinations [3]. Unfortunately, these treatment methods are inextricably linked with many side effects, such as pain, emotional stress, fatigue, a negative impact on fertility, and subsequent cancers [4]. Biologically active molecules in medicinal plants can be employed to reduce side effects and support the efficacy of the therapy. Notably, *Potentilla* species are widely used in traditional medicine for the treatment of dysentery, diarrhoea, diabetes mellitus, unspecified forms of cancer, and inflammation of the skin [5,6]. The pharmacological properties of *Potentilla* species stem from their secondary metabolite composition, which includes a predominant presence of polyphenols, such as hydrolysable and condensed tannins, flavonoids, and phenolic acid, as well as triterpenoids. These substances are associated with antioxidant, anti-inflammatory, and antimicrobial properties [5]. Numerous in vitro experiments on compounds obtained from *Potentilla* species have shown efficacy against various cancer cell lines, e.g., methanol extract from *P. discolor* inhibited the proliferation and induced the apoptosis of MC3 and YD-15 (human mucoepidermoid carcinoma) [7], ethyl acetate extracts from *P. recta* and *P. astracanica* decreased viability of HEp-2 (human cervix carcinoma) [8], and selected extracts and fractions from aboveground materials of *P. alba* significantly reduced the viability and proliferation of HT-29 (human colon adenocarcinoma) [9]. In a previous study, we demonstrated that aqueous acetone extracts from the aerial parts of selected *Potentilla* species showed great chemopreventive potential by decreasing the viability and proliferation of LS180 (human colon adenocarcinoma) cells, simultaneously causing substantial damage to their cell membranes while having a significantly weaker impact on normal colon epithelial cell line CCD841 CoN [10]. The present study is a continuation of that previous investigation conducted by the authors, concerning an assessment of the cytotoxicity and antiproliferative effect of aqueous acetone extracts from selected, rare *Potentilla* species against human colon cancer cell line LS180 and normal colon epithelial cell line CCD841 CoN. Additionally, identification of the marker metabolites present in extracts using LC–HRMS analysis was conducted to reveal and validate correlations between the qualitative chemical composition of the investigated samples and possible mechanisms of action.

## 2. Results and Discussion

### 2.1. Determination of Total Secondary Metabolites Content

Polyphenols are among the major secondary metabolites that are accountable for the pharmacological activities of plant-based preparations. The major group of polyphenols include flavonoids, phenolic acids, hydrolysable and condensed tannins, lignans, and stilbenes [11]. *Potentilla* species are well-known for their abundance of tannins and flavonoids, which contribute to certain traditional applications aimed at tackling diarrhoea, microbial infections, inflammations of the upper and lower gastrointestinal tract, diabetes mellitus, etc. [5,12]. In our study, extracts from the aerial and underground parts of common and rare *Potentilla* species were prepared using 70% acetone and were quantitative assessed for the general polyphenolic classes contents using colourimetric methods. The level of phenolic compounds in the extracts from selected *Potentilla* species are presented in Table 1. Extracts from the underground parts, namely, **PAL7r** and **PER7r**, were found to contain the highest total phenolic (TPC) and total tannin (TTC) contents (268.63, 237.56, and 326.28, 269.79 mg gallic acid equivalent (GAE)/g extract, respectively). On the other hand, among extracts from the aerial parts, **PFR7** and **PPE7** had the highest TPC and TTC values (240.1, 178.65, and 218.85, 195.97 mg GAE/g extract, respectively), while **PAU7** and **PTH7** revealed the lowest TPC and TTC values (148.38, 129.2, and 149.77, 132.55 mg GAE/g extract, respectively). Moreover, **PFR7** was found to contain the highest total proanthocyanidin content (TPrC) (53.59 mg catechin equivalent (CE)/g extract), notably higher than that of other herb extracts. According to our previous study and the results herein, extracts from rhizomes, namely, **PAL7r** and **PER7r**, had remarkably higher proanthocyanidin contents than their above-ground counterparts (72.63 and 61.61 vs. 21.28 and 2.05 mg CE/g extract, respectively) [12]. Moreover, **PAL7r** and **PER7r** had the highest total phenolic acid content (TPAC), followed by **PFR7** (221.08, 263.54, and 197.83 mg caffeic acid equivalent (CAE)/g extract, respectively). On the contrary, **PAL7r** and **PER7r** had the lowest total flavonoid content (TFC) values, which were significantly lower than those of all other extracts. **PHY7** and **PPE7** revealed the highest TFC values (113.29 and 108.2 mg rutin equivalent (RE)/g extract, respectively). All the obtained results were significantly higher than the values available in the literature data reported for various extracts from the aerial parts of *P. erecta*, *P. fruticosa*, *P. nepalensis*, *P. pensylvanica*, and *P. thuringiaca* [13,14,15]. Notably, the selection of the solvent in the extraction process is a crucial factor in the explanation of those differences. An aqueous acetone solvent extracts much fewer non-phenol compounds, such as carbohydrates, than methanol and water, which results in higher TPC and TFC values [16]. Moreover, aqueous acetone was reported as an excellent solvent for extracting higher molecular weight flavonoids and proanthocyanidins [17]. The aforementioned solvent prevents the decomposition of hydrolysable tannins during the extraction process, leading to a higher tannin content in the obtained extracts [18].

### 2.2. LC–HRMS Qualitative Analysis of Selected Extracts

The identification of the secondary metabolite composition of the aqueous acetone extracts of selected *Potentilla* species using LC–HRMS (liquid chromatography–high-resolution mass spectrometry) analysis demonstrated the presence of 198 compounds. Among them, three groups of phenolic compounds were dominant in the analysed extracts: tannins, flavonoids, and phenolic acids. Monomeric and dimeric ellagitannins, such as agrimoniin, sanguinis and pedunculagin, are important chemophenetic markers in the Rosaceae family, especially in the *Potentilla*, *Rubus*, and *Fragaria* genera [19]. The chromatographic analysis reported herein led to the identification of a series of hydrolysable tannins that are represented by ellagitannin derivatives, such as laevigatin isomers (**84**, **109**, **114**, **124**, and **128**), laevigatin E isomers (**37** and **40**), agrimoniin (**162**) and its structural isomer (**151**), agrimonic acid A or B (**102**), galloyl-HHDP-glucose (**16**, **21**, **43**, and **48**), digalloyl-HHDP-glucose (**33** and **60**) and trigalloyl-HHDP-glucose (**131** and **133**), galloyl-bis-HHDP-glucose (**108**, **118**, and **144**), ellagic acid (**135**) and its *O*-pentosides (**132** and **163**), *O*-hexosides (**73**, **97**, and **101**), and uronic acid (**82**, **95**, and **130**) derivatives. The analysis indicated that the one of the most abundant phytochemicals in all the extracts, except **PAL7r**, was agrimoniin. Agrimoniin has been frequently described as the major phenolic compound in several *Potentilla* species, such as *P. argentea*, *P. anserina*, *P. grandiflora P. kleiniana P. norvegica, P. recta*, and *P. rupestris* [10,20,21,22]. Other present ellagitannins, namely, leavigatins and agrimonic acid, are formed from the partial hydrolysis of agrimoniin (dehydrodigalloyl-di-(bis-HHDP-glucose)) [23]. Furthermore, few degradation products of hydrolysable tannins degradation, such as ellagic acid (**135**), brevifolincarboxylic acid (**46**) and its structural isomer (**50**), and brevifolin (**83**), were found. Gallotannins were present in a few extracts, which showed the presence of di-, tri-, tetra-, and pentagalloylglucose isomers (**35**, **36**, **80**, **86**, **103**, **137**, and **168**). However, the analysis revealed the absence of hydrolysable tannins in **PAL7r**. These findings are in agreement with the previous study, which demonstrated the absence of these metabolites in the aerial parts of *P. alba* [9]. Moreover, the analysis revealed the presence of condensed tannins, especially in **PAL7r**, such as catechin (**28**), epicatechin (**61**), and their glucosides (**11**, **22**, **23**, **41**, and **106**), as well as products of their polymerisation, such as A-type procyanidins (**24**, **54**, **71**, **90**, **96**, and **110**) and dimeric (**66**, **88**, and **107**), trimeric (**7**, **42**, **45**, and **64**) and tetrameric (**56** and **93**) B-procyanidins, including procyanidin B1 (**25**), procyanidin B2 (**47**), procyanidin B3 (**27**), procyanidin C1 (**94**), and procyanidin C2 (**34**).

Based on the chromatographic profiles, a number of flavonoids were detected and characterised, including apigenin (**92**, **119**, **161**, **166**, **184**, and **185**) as well as isorhamnetin (**87**, **91**, **98**, **100**, **125**, **150**, **158**, **167**, **169**, **171–173**, **179**, **182**, **187**, **188**, **191**, **194**, and **196**), naringenin (**180**), kaempferol (**62**, **67**, **72**, **78**, **81**, **104**, **113**, **117**, **140**, **142**, **143**, **147**, **148**, **153**, **155**, **157**, **159**, **160**, **164**, **165**, **170**, **175**, **176**, **178**, **190**, **192**, and **193**), quercetin (**39**, **44**, **51**, **53**, **55**, **58**, **59**, **63**, **74**, **79**, **85**, **99**, **105**, **111**, **116**, **120–122**, **126**, **127**, **129**, **134**, **136**, **138**, **139**, **141**, **145**, **149**, **152**, **154**, **156**, **177**, and **186**), acacetin (**183**), and tricin (**189** and **195**) derivatives. From a chemophenetic perspective, a few of them may be useful as chemical markers of the *Potentilla* genus, such as both isomers of tiliroside (**190**), astragalin (kaempferol 3-*O*-glucoside) (**155**), isorhamnetin 3-*O*-glucoside (**169**), kaempferol 3-*O*-glucuronide (**157**), avicularin (quercetin 3-*O*-arabinoside) (**149**), hyperoside (quercetin 3-*O*-galactoside) (**139**), isoquercitrin (quercetin 3-*O*-glucoside) (**136**), and rutin (quercetin 3-*O*-rutinoside) (**138**), which were previously reported to be present in at least one of the *Potentilla* species investigated to date [5,6,10]. The analysis also revealed the presence of phenolic acids, such as gallic acid (**1**), caffeic acid (**29**) and its derivatives (**5**, **9**, **10**, **15**, **22**, **65**, and **75**), coumaric acid (**12**, **17**, **57**, and **197**), dihydroxybenzoic acid (**13**)**,** and syringic acid (**89**) derivatives. The detailed chromatographic data of the analysed samples are shown in Table 2 and in Appendix A. To summarize, the number of compounds shared by all the analysed *Potentilla* species may typify their chemical profile as homogeneous.

### 2.3. Examination of the Anticancer Potential of Extracts

In the first step, the extract’s influence on both human colon epithelial cell line CCD841 CoN as well as human colon adenocarcinoma cell line LS180 was examined using an MTT assay. Studies were conducted after 48 h of the cells being exposed to either a culture medium (control) or extracts (25–250 µg/mL). As presented in Figure 1 and Table 3, all the investigated extracts inhibited the metabolic activity of both normal and cancer cells, and the observed effect was dose-dependent. The most significant anticancer effect was presented by extracts **PAL7r** and **PFR7**, which, at the highest tested concentration, deceased LS180 cells’ proliferation by 91.3% (IC_50 PAL7r LS180_ = 82 µg/mL) and 94.8% (IC_50 PFR7 LS180_ = 89 µg/mL), respectively. On the contrary, the weakest influence on the metabolic activity of colon cancer cells was noted after treatment with **PTH7** and **PRI7**, which, at a concentration of 250 µg/mL, inhibited cell viability by 58.7% (IC_50 PTH7 LS180_ = 225 µg/mL) and 57.9% (IC_50 PRI7 LS180_ = 213 µg/mL), respectively. The strongest reduction (by 36.7%) of the viability of colon epithelial cells was caused by both **PME7** and **PHY7** (IC_50 PME7 CCD841 CoN_ = 380 µg/mL; IC_50 PHY7 CCD841_ CoN = 489 µg/mL), while the weakest effect, as reflected by the IC_50_ value, was shown by PRI7 (IC_50 PRI7 CCD841 CoN_ = 2402 µg/mL).

Used as a positive control for the experiment, 5-fluorouracil (5-FU) at a concentration of 25 µM decreased the metabolic activity of CCD841 CoN and LS180 by 22.2% and 46.2%, respectively. All the investigated extracts at the highest tested concentrations revealed a stronger anticancer effect than 5-FU. Seven of twelve extracts inhibited LS180 cells’ viability better than 5-FU, when used at lower concentrations; the beneficial effect of **PER7r**, **PHY7**, **PME7**, and **PPE7** was observed at concentrations of 150 and 250 µg/mL, while the beneficial effect of **PAL7r**, **PFR7**, and **PNE7** was observed at concentrations from 100 to 250 µg/mL. In the case of CCD841 CoN cells, only 3 out of 12 of the investigated extracts inhibited the metabolic activity of colon epithelial cells stronger than 25 µM 5-FU: **PAL7r** (250 µg/mL); **PHY7** (250 µg/mL); **PME7** (150 and 250 µg/mL).

The obtained results for the MTT assay may be strongly associated with high TPrC, especially in the **PAL7r**, **PER7r**, and **FFR7** extracts. On several occasions, proanthocyanidins were reported to have a strong influence on colon cancer cell viability. Especially oligomeric proanthocyanidins from grape seeds (*Vitis vinifera* L., Vitaceae), which induce the apoptotic cell death of Caco-2 (human colorectal adenocarcinoma) cells manifested by nuclear condensation, caspase-3 and PARP cleavage, and formation of apoptotic bodies [31]. Additionally, proanthocyanidins from hops (*Humulus lupulus* L., Cannabaceae) increased the intracellular formation of reactive oxidative species, which was manifested by the augmented accumulation of protein carbonyls and induced cytoskeletal disorganisation of human colon cancer cell line HT-29 [32]. However, in a comparison with a previous study, all extracts exerted a weaker effect on cancer cell viability than extracts obtained from five out of six tested aqueous acetone extracts, namely, *P. argentea*, *P. grandiflora*, *P. norvegica*, *P. recta*, and *P. rupestris* [10]. This difference may be associated with the lower TPC and TTC obtained herein. Ellagitannins display great chemopreventive and chemotherapeutic activities. Among them, agrimoniin, the main ellagitannin present in all extracts, except **PAL7r**, was shown to have prominent anticancer, antioxidant, and anti-inflammatory activities [33]. It is widely recognised that there is a strict correlation between chronic inflammation and colorectal cancerogenesis [34]. Preclinical and clinical studies showed that non-steroidal and anti-inflammatory drugs are effective in preventing the formation of colorectal tumours; however, there are limitations due to severe and fatal side effects, such as gastric bleeding, ulcers, and renal toxicity [35]. Phytochemicals have fewer side effects compared with synthetic drugs, which is advantageous. A study conducted by Shi and co-authors revealed that the use of lyophilised strawberries (*Fragaria x ananasa* L., Rosaceae) containing agrimoniin as the second-most-abundant phytochemical, in an inflammation-induced colorectal carcinogenesis model, led to downregulating the mRNA expression of the proinflammatory mediators, such as COX-2, iNOS, IL-1β, IL-6, and TNF-α [36]. Moreover, in two consecutive studies, an agrimoniin-enriched fraction from the underground parts of *P. erecta* showed the dose-dependent inhibition of UVB-induced or TNF-α stimulated IL-6 and PGE_2_ production as well as reduced NFκB activation in HaCaT cells (human keratinocytes). Further, a UV erythema study in healthy volunteers revealed that an agrimoniin-enriched fraction significantly inhibited the UVB-induced inflammation process [37,38].

In the next step, the antiproliferative activity of *Potentilla* L. extracts was examined in both the normal and cancer cell lines using a BrdU assay (Figure 2 and Table 1). All the investigated extracts decreased DNA synthesis in the colon cancer cells in a dose-dependent manner. Nevertheless, a significant decrease in LS180 cells’ proliferation in response to the extract, for the whole range of tested concentrations, was only observed for **PAL7r**, **PFR7**, and **PER7r**, which, at concentrations of 100, 150, and 250 µg/mL, reduced DNA synthesis by around 80%. Furthermore, the aforementioned extracts were characterised by the lowest IC_50_ values (IC_50 PAL7r LS180_ = 52 µg/mL; IC_50 PFR7 LS180_ = 50 µg/mL; IC_50 PER7r LS180_ = 54 µg/mL). On the contrary, the lowest antiproliferative abilities were revealed by PAU7, which, even at the highest tested concentration, decreased DNA synthesis in LS180 cells by only 14.9% (IC_50 PAU7 LS180_ = 1495 µg/mL). The antiproliferative effect of the examined extracts was also observed in colon epithelial cells; however, the observed effect was weaker than in cancer cells. The only extract that did not affect divisions of CCD841 CoN was **PER7**, which was characterised by the highest IC_50_ value of 3705 µg/mL. On the contrary, the most significant changes in normal cells were observed in response to **PAL7r**, **PFR7**, and **PER7r**, which, at the highest tested concentration, reduced the proliferation of epithelial cells by 36.1%, 38.3%, and 43.9%, respectively (IC_50 PAL7r CCD841 CoN_ = 412 µg/mL; IC_50 PER7 CCD841 CoN_ = 282µg/mL; IC_50 PER7r CCD841 CoN_ = 337 µg/mL). As presented in Figure 2, 25 µM 5-fluorouracil (5-FU) decreased DNA synthesis in the investigated cell lines to 90.7% (CCD841 CoN) and 29.7% (LS180). The antiproliferative effect of 5-FU recorded in colon cancer cells was significantly stronger than the changes induced by most of the examined extracts (9 of 12); however, **PAL7r**, **PFR7**, and **PER7r**, in concentrations from 100 to 250 µg/mL, decreased DNA synthesis more than 5-FU. On the contrary, data collected from colon epithelial cells revealed that most of the investigated extracts (**PAL7r**, **PER7r**, **PFR7**, **PHY7**, **PME7**, **PNE7**, **PPE7**, **PPU7**, **PRI7**, and **PTH7**) at higher concentrations inhibited DNA synthesis stronger than 25 µM 5-FU, while the antiproliferative effect of **PFR7** for the whole range of tested concentrations was higher than the changes induced by analysed cytostatic. However, the presented results correspond with data from our previous study, showing that tested *Potentilla* species possess similar anticancer potential; moreover, for the **PAL7r**, **PER7r**, and **PFR7** extracts, the results from a BrdU assay were significantly higher than those for all other tested samples [10]. The observed effect may be attributed to the high TPrC values. Kresty and co-authors found that a cranberry (*Vaccinium macrocarpon* Aiton, Ericaceae) proanthocyanidin-rich fraction significantly inhibited the viability and proliferation of human oesophageal adenocarcinoma SEG-1 cells. The mechanism involved cell cycle arrest in the G1 phase as well as a significant apoptosis induction [39]. Notably, the antiproliferative effect of other extracts may be connected with the presence of ellagitannins and the main product of their decomposition, namely, ellagic acid. The anticancer mechanism of ellagic acid is multidirectional. A study conducted on human colorectal carcinoma HCT-116 and the Caco-2 cell line revealed that ellagic acid induced cell cycle arrest in the G phase, reduced proliferating cell nuclear antigen (PCNA) expression and mitotic activity, and induced apoptosis via increasing the expression of caspase-8 and Bax [40]. Additionally, a further study conducted on HCT-116 cells revealed the involvement of ellagic acid in the decreased gene expression of signalling pathways’ proteins such as mitogen-activated protein (MAPK), p53, PI3K-Akt, and TGF-β [41]. Recently, Han and co-authors found that tiliroside acted as an inhibitor of carbonic anhydrase XII (CAXII), a membrane enzyme that produces a favourable intracellular pH and sustains optimum P-glycoprotein (Pgp) efflux activity in cancer cells. Moreover, tiliroside downregulated E2F1 and E2F3 expression and promoted caspase-3 activity [39]. In addition, the meta-analysis revealed that a high intake of flavonoids, such as quercetin and kaempferol derivatives, in the diet may decrease the risk of colon cancer [42].

Extract cytotoxicity was also examined in both normal and cancer colon cell lines using an LDH (lactate dehydrogenase)-based assay (Figure 3). Most of the examined extracts were not cytotoxic against human colon epithelial cells; however, **PME7** was, for the whole range of tested concentrations, while **PAL7r**, **PER7**, and **PHY7**, in concentrations from 100 to 250 µg/mL, damaged the membranes of epithelial cells. The indicated extracts at the highest tested concentration increased the LDH level by an average of 11%. Studies conducted on colon cancer cells showed the cytotoxic effect of all the examined extracts for the whole range of tested concentrations. The strongest damage of cancer cell membranes was caused by **PAL7r**, which in concentrations from 25 to 250 µg/mL increased the LDH level by 145.7% and 479.0% respectively. The weakest cytotoxic effect was noted in colon cancer cells treated with **PRI7**, **PPU7**, and **PTH7**, which, at the highest tested concentration, caused an increase in the LDH level by 245.1%, 254.7%, and 256.0%, respectively. An LDH assay showed that 5-FU in a concentration of 25 µM was not cytotoxic against colon epithelial, while LDH releases were increased in colon cancer cells by 13.4%. All the investigated extracts damaged the colon cancer cell membranes significantly more than 5-FU. For CCD841 CoN cells, significant differences in the LDH concentration between 25 µM 5-FU and the examined extracts were observed in the case of four extracts (**PME7**, **PAL7r**, **PER7**, and **PHY7**), for which the cytotoxic impact on colon epithelial cells was reported above. The results of the LDH assay are presumably directly associated with the TTC in the investigated samples. Tannins, due to their specific chemical structure, are known to affect the physical properties of membranes, initiate membrane protein aggregation, increase bilayer adhesion, and regulate cell metabolism [43,44]. The most abundant hydrolysable tannin present in most extracts, agrimoniin, induces the intrinsic pathway of apoptosis, directly influencing the permeability of the mitochondrial membrane via the activation of the mitochondrial permeability transition pore (MPTP) [45]. However, further in vivo studies are required to evaluate the exact mechanism of action. The bioavailability of large ellagitannins is generally low. Therefore, the method of application is limited to topical application. The gut microbiota metabolise ellagitannins and ellagic acid to produce a series of bioavailable metabolites, known as urolithins. Urolithins possess a series of biological activities, such as anti-inflammatory, antioxidant, anticancer, and immunomodulatory activities. The chemopreventive effects of urolithins were extensively studied in several models, including prostate and colorectal cancer models. Urolithins were shown to inhibit colon cancer cell growth in a dose-dependent manner, alter the expression of the genes and proteins modulating the cell cycle, and induce apoptosis [46]. Notably, a clinical study on the aerial parts of *P. anserina* and the rhizomes of *P. erecta* confirmed the formation of urolithins in ex vivo conditions [47].

## 3. Materials and Methods

### 3.1. Chemicals

The reference phytochemicals, including isorhamnetin 3-*O*-glucoside, kaempferol 3-*O*-glucuronide, and quercetin 3-*O*-glucuronide were obtained from Extrasynthese (Genay, France). Gallic acid, catechin, and epicatechin were obtained from Carl Roth (Karlsruhe, Germany). Procyanidin B1, procyanidin B2, procyanidin B3, and procyanidin C1 were purchased from Cayman Chemical (Ann Arbor, MI, USA), while agrimoniin, apigenin, 3-*O*-caffeoylquinic acid, ellagic acid, astragalin (kaempferol 3-*O*-glucoside), pedunculagin, avicularin (quercetin 3-*O*-arabinoside), hyperoside (quercetin 3-*O*-galactoside), isoquercitrin (quercetin 3-*O*-glucoside), and tiliroside (purity > 96%) were previously isolated in the Department of Pharmacognosy of Medical University of Białystok (Białystok, Poland) [22,48,49,50,51]. All other analytical grade chemicals used in the study were obtained from Sigma-Aldrich (St. Louis, MO, USA). To obtain ultra-pure water, a POLWATER DL3-100 Labopol (Kraków, Poland) system was used. Investigated extracts (100 mg/mL) and 5-fluorouracil (50 mM) were dissolved in dimethyl sulfoxide (DMSO) to prepare stock solutions. Working solutions were prepared by dissolving stock solutions in a culture medium. The final concentration of DMSO in all working solutions used in the studies was 0.25%.

### 3.2. Plant Materials and Extraction Procedure

Plants used to obtain material for investigations come from the Medicinal Plant Garden at the Medical University of Białystok (Białystok, Poland) and were collected in June-August 2017–2020. Plants were carefully identified by one of the authors (M.T.), and individual voucher specimens were deposed at the Herbarium of the Department of Pharmacognosy, Medical University of Białystok (Białystok, Poland). Plant material was dried at room temperature in the shade and air temperature and subsequently finely ground with an electric grinder. Accurately weighed 2 g of each powdered dry plant material were separately extracted using an ultrasonic bath (Sonic-5, Polsonic, Warszawa, Poland) with 70% acetone at a controlled temperature (40 ± 2 °C) for 45 min in a 1:75 (*w*:*v*) solvent ratio to obtain raw extracts. Subsequently. extracts were evaporated to dryness, diluted with water (50 mL). and successively portioned between chloroform (10 × 20 mL). Afterwards. purified extracts were freeze-dried. The list of obtained aqueous acetone extracts from selected *Potentilla* species detailing plant species, voucher specimen, the parts used and extraction yields are presented in Table 4.

### 3.3. Phytochemical Profile

#### 3.3.1. Determination of Total Phenolic (TPC) and Total Tannin Content (TTC)

The content of total phenolic compounds was measured using standard the Folin-Ciocalteu colourimetric method, with slight modification according to [29]. The content total tannin determination was carried out using the hide powder-binding method and Folin–Ciocalteu assay reported in the corresponding monograph in the European Pharmacopoeia 10th ed. [52]. The absorbance was measured at 760 nm using a EPOCH2 BioTech (Winooski, VT, USA) microplate reader. The obtained results for were expressed as milligrams of gallic acid equivalents per gram of extract (mg GAE/g extract). The determination was repeated at least in triplicate for each sample solution.

#### 3.3.2. Determination of Total Proanthocyanidin Content (TPrC)

Total proanthocyanidin content was analysed using the procedure based on the previously published protocol [53]. The analysis was carried out by mixing 50 µL of the sample solution (1 mg/mL) dissolved in methanol and 250 µL of 0.1% methanolic solution of 4-dimethylamino-cinnamaldehyde (DMCA) reagent in 6M HCl. After incubation of the mixture at room temperature for 15 min, the absorbance was measured at 635 nm, and results were expressed as milligrams of catechin equivalents per gram of extract (mg CE/g extract). The determination was repeated at least five times for each sample solution.

#### 3.3.3. Determination of Total Phenolic Acid Content (TPAC)

Total phenolic acid content was estimated with the procedure using Arnov’s reagent (1 g of sodium molybdate and 1 g of sodium nitrate dissolved in 10 mL of distilled water) [54]. Each time the tested solution (30 µL) was mixed with 180 μL of water, 30 μL of 0.5 M HCl, 30 μL of Arnov’s reagent, and 30 μL of 1 M NaOH were sequentially added to the microplate well, and then it was incubated for 10 min at ambient temperature. Afterwards, the absorbance was measured at 490 nm, and results were expressed as milligrams of caffeic acid equivalents per gram of extract (mg CAE/g extract). The determination was repeated at least three times for each sample solution.

#### 3.3.4. Determination of Total Flavonoid Content (TFC)

Total flavonoid content was estimated by the previously described colourimetric method [29]. Each aliquot (100 µL) was mixed with aluminum chloride (AlCl_3_) solution (100 µL, 2% *w*:*v*). After incubation of the mixture at room temperature for 10 min, the absorbance was measured at 415 nm, and results were expressed as milligrams of rutin equivalents per gram of extract (mg RE/g extract). The determination was repeated at least three times for each sample solution.

#### 3.3.5. LC–HRMS Profiling of Extracts

The separation and qualitative evaluation of each extract were conducted using a Kinetex XB-C18 column (150 × 2.1 mm, 1.7 μm, Phenomenex, Torrance, CA, USA) and Agilent 1260 Infinity LC chromatography system coupled to a photo-diode array (PDA) detector and 6230 time-of-flight (TOF) mass spectrometer (Santa Clara, CA, USA). A detailed description of the execution of the above-mentioned assay was presented in the previous study [10].

### 3.4. Cell Cultures

For the cell culture study, human colon adenocarcinoma cell line LS180 and human colonic epithelial cell line CCD841 CoN were purchased from the European Collection of Cell Cultures (ECACC, Centre for Applied Microbiology and Research, Salisbury, UK) and American Type Culture Collection (ATCC, Menassas, VA, USA), respectively. LS180 cells and CCD841 CoN cells were maintained in Dulbecco′s Modified Eagle′s Medium/Nutrient Mixture F-12 Ham and Dulbecco’s Modified Eagle’s Medium (DMEM), respectively. Then, 10% fetal bovine serum (FBS), penicillin (100 U/mL), and streptomycin (100 g/mL) were added to the cell culture media. Cells were incubated in a humidified atmosphere of 95% air and 5% CO_2_ at 37 °C.

### 3.5. Evaluation of the Anticancer Potential of Extracts

Examination of the anticancer potential of extracts was conducted simultaneously on both cancer (LS180) and normal (CCD841 CoN) colon cells. Cells at a density of 5 × 104 cells/mL were plated on 96-well plates. The next day, the cell growth medium was exchanged for fresh medium supplemented with investigated extracts or 25 μM 5-fluorouracil (5-FU). After 48 h of treatment, compound impacts on cell membrane integrity, metabolic activity, and DNA syntexis were examined using LDH, MTT, and BrdU assays, respectively. The description of the execution of indicated tests was previously presented [10].

### 3.6. Statistical Analysis

The data were presented as the mean ± standard error of mean (SEM). Statistical analyses were carried out using one-way ANOVA with Tukey’s post hoc test and column statistics. Significance was accepted at *p* < 0.05. The IC_50_ value (concentration causing proliferation inhibition by 50% compared to control) was calculated according to the method of Litchfield and Wilcoxon [55] using GraphPad Prism 5.

## 4. Conclusions

In conclusion, the presented study reports, for the first time, an analysis of the LC–HRMS profile of aqueous acetone extracts from rare *Potentilla* species. The analysis revealed a series of marker metabolites such as agrimoniin, pedunculagin, dimeric and trimeric B-type procyanidins, tiliroside, astragalin (kaempferol 3-*O*-glucoside), hyperoside (quercetin 3-*O*-galactoside, ellagic acid, and tri-coumaroyl spermidine. The performed studies revealed that all of the investigated acetone extracts obtained from rare *Potentilla* species decreased the viability and proliferation of human colon adenocarcinoma LS180 cells. Nevertheless, most of the investigated extracts also decreased metabolic activity and DNA synthesis in human colon epithelial CCD841 CoN cells, and 4 out of 12 of the tested extracts (**PAL7r**, **PER7**, **PHY7**, and **PME7**) showed cytotoxic effects against normal epithelial cells. Despite the fact that the investigated extracts affected both normal and cancer colon cells, the LS180 cells were more sensitive to tested extracts. Considering the data obtained from all the performed studies, the 2 of the 12 investigated extracts (**PFR7** and **PER7r**) revealed the greatest chemopreventive potential, as manifested by the effective elimination of colon cancer cells, which caused both damage to their cell membranes and inhibition of their proliferation and metabolic activity, with a simultaneous lack of any cytotoxic effect on normal colon epithelial cells and a significantly weaker effect on their metabolism and DNA synthesis compared to cancer cells. The previous [10] and currently obtained results indicated that some acetone extracts from *Potentilla* species have anticancer potential, however, additional animal and clinical studies, especially including the influence of intestinal flora are required to verify discovered beneficial properties of investigated extracts. Nevertheless, discovered selectivity of the anticancer effects of tested extracts encourages further studies to develop a new efficient and safe therapeutic strategy for people who have been threatened by or suffered from colon cancer.

## Figures and Tables

**Figure 1 ijms-24-04836-f001:**
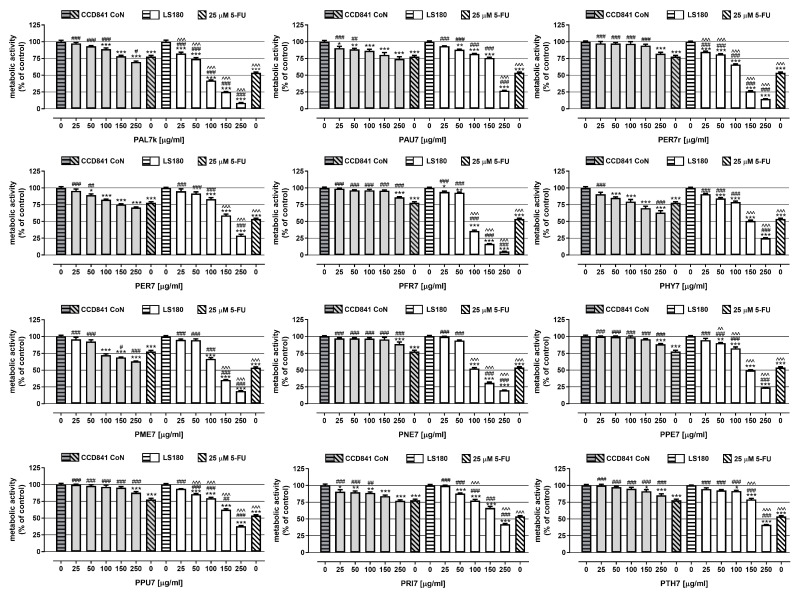
Influence of obtained extracts on the viability of CCD841 CoN and LS180 cell lines. The cells were exposed for 48 h to the culture medium alone (control). extract at concentrations of 25–250 µg/mL. or 25 μM 5-fluorouracil (5-FU; positive control). Metabolic activity of investigated cell lines in response to tested compounds was examined photometrically by MTT assay. Results are presented as mean ± SEM of at least 5 measurements. * *p* < 0.05; ** *p* < 0.01; *** *p* < 0.001 vs. control. # *p* < 0.05; ## *p* < 0.01; ### *p* < 0.001 vs. positive control. ^^ *p* < 0.01; ^^^ *p* < 0.001 colon cancer cells treated with extract/5-FU vs. colon epithelial cells exposed to the extract/5-FU at the corresponded concentration; one-way ANOVA test; post-test: Tukey’s.

**Figure 2 ijms-24-04836-f002:**
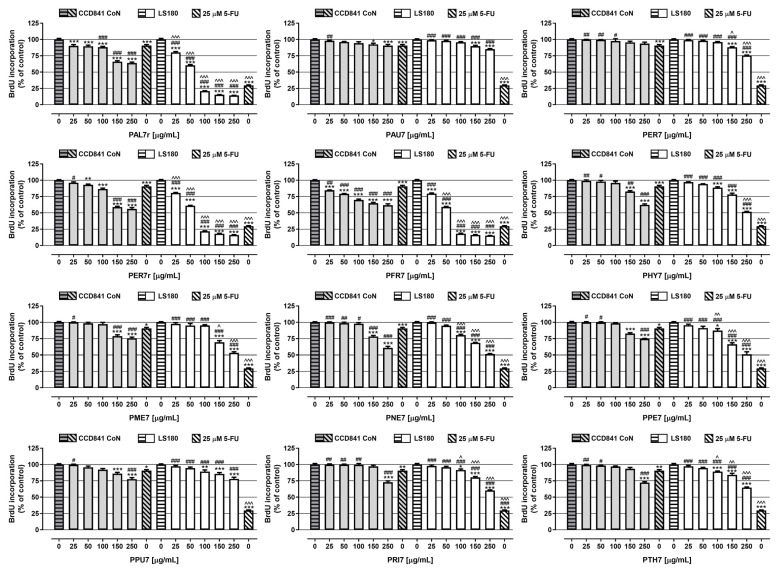
Antiproliferative effect of extracts on CCD841 CoN and LS180 cell lines. The cells were exposed for 48 h to the culture medium alone (control). extract at concentrations of 25–250 µg/mL. or 25 μM 5-fluorouracil (5-FU; positive control). The antiproliferative impact of investigated compounds was measured using BrdU assay (incorporation of BrdU to newly synthesised DNA). Results are presented as mean ± SEM of at least 4 measurements. * *p* < 0.05; ** *p* < 0.01; *** *p* < 0.001 vs. control. # *p* < 0.05; ## *p* < 0.01; ### *p* < 0.001 vs. positive control. ^ *p* < 0.05; ^^ *p* < 0.01; ^^^ *p* < 0.001 colon cancer cells treated with extract/5-FU vs. colon epithelial cells exposed to the extract/5-FU at the corresponded concentration; one-way ANOVA test; post-test: Tukey’s.

**Figure 3 ijms-24-04836-f003:**
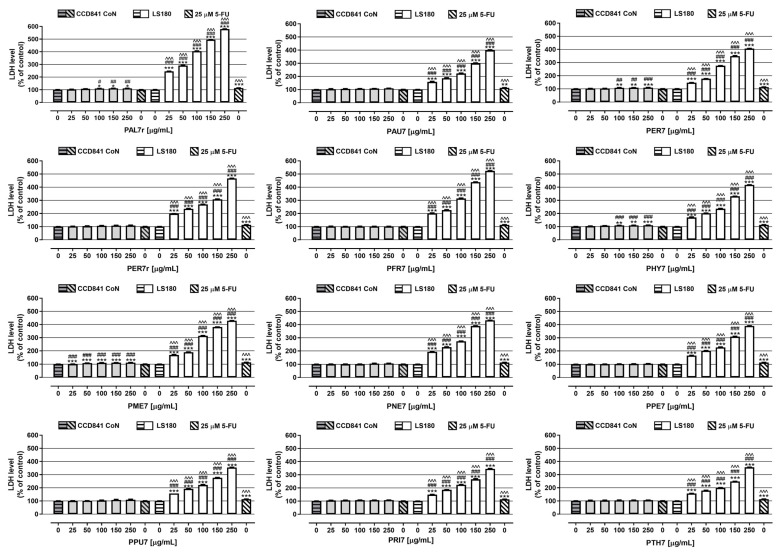
The influence of extracts on cell membrane integrity of CCD841 CoN and LS180 cell lines. The cells were exposed for 48 h to the culture medium alone (control). extract at concentrations of 25–250 µg/mL. or 25 μM 5-fluorouracil (5-FU; positive control). Extracts’ cytotoxicity (level of LDH released into the cell culture medium from damaged cell membranes) was measured using an LDH assay. Results are presented as mean ± SEM of at least 3 measurements. * *p* < 0.05; ** *p* < 0.01; *** *p* < 0.001 vs. control. # *p* < 0.05; ## *p* < 0.01; ### *p* < 0.001 vs. positive control. ^^^ *p* < 0.001 colon cancer cells treated with extract/5-FU vs. colon epithelial cells exposed to the extract/5-FU at the corresponded concentration; one-way ANOVA test; post-test: Tukey’s.

**Table 1 ijms-24-04836-t001:** Total phenolic (TPC), tannin (TTC), proanthocyanidin (TPrC), phenolic acid (TPAC), and flavonoid contents (TFC) of aqueous acetone extracts.

Samples	TPC (mg GAE/g Extract) ^1^	TTC (mg GAE/g Extract) ^1^	TPrC (mg CE/g Extract) ^2^	TPAC (mg CAE/g Extract) ^3^	TFC (mg RE/g Extract) ^4^
**PAL7r**	268.6 ± 6.9	237.6 ± 5.7	72.6 ± 2.5	221.1 ± 7	15 ± 0.3
**PAU7**	148.4 ± 2.3	129.2 ± 2	3.4 ± 0.1	44.2 ± 1.4	59.7 ± 1.3
**PER7**	201.2 ± 4.3	169.2 ± 7	2.1 ± 0.1	59.9 ± 1.3	54.9 ± 0.4
**PER7r**	326.3 ± 3.5	269.8 ± 2.4	61.6 ± 1.1	263.5 ± 7.5	11 ± 0.1
**PFR7**	240.1 ± 6.1	178.7 ± 5.5	53.6 ± 0.9	197.8 ± 6.2	94.6 ± 2.4
**PHY7**	199.2 ± 1.7	178.2 ± 3.9	1.6 ± 0.1	44 ± 1.1	113.3 ± 1.5
**PME7**	195.3 ± 4.4	168.5 ± 3.6	13.1 ± 0.4	80.8 ± 2	84.6 ± 0.1
**PNE7**	188.8 ± 2.5	163.5 ± 0.5	1.1 ± 0.1	33.4 ± 0.3	66.5 ± 2.5
**PPE7**	218.9 ± 1.8	196 ± 3.1	0.2 ± 0.1	50.5 ± 0.5	108.2 ± 0.5
**PPU7**	151.5 ± 2.4	135.9 ± 2.4	5.5 ± 0.1	50.2 ± 2.1	64.9 ± 0.6
**PRI7**	212.2 ± 5.5	170.5 ± 4.4	5.6 ± 0.6	58.1 ± 1.7	84.4 ± 0.7
**PTH7**	149.8 ± 2.3	132.6 ± 2.3	4.9 ± 0.1	58.8 ± 2.5	76.4 ± 1.6

^1^ GAE—gallic acid equivalent; ^2^ CE—catechin equivalent; ^3^ CAE—caffeic acid equivalent; ^4^ RE—rutin equivalent. All values represent the mean ± standard deviation of three replicates for each sample (n = 3).

**Table 2 ijms-24-04836-t002:** LC–HRMS qualitative analysis of aqueous acetone extracts from aerial and underground parts of selected *Potentilla* species.

No.	Compounds	Rt (min)	UV Spectra (λ Max nm)	Observed ^1^	Δ (ppm)	Formula	Fragmentation		Presence in Extracts												Ref.
							Negative	Positive	PAL7r	PAU7	PER7	PER7r	PFR7	PHY7	PME7	PNE7	PPE7	PPU7	PRI7	PTH7	
1	Gallic acid	5.72	270	169.01335	−3.02	C_7_H_6_O_5_	**169**, 125			+	+	+	+	+	+	+	+	+	+	+	(s)
2	2-Pyrone-4,6-dicarboxylic acid	6.82	316	182.99292	−2.56	C_7_H_4_O_6_	366, **183**, 139	**185**		+	+	+	+		+	+	+	+	+	+	[24]
3	Bis-HHDP-gluconic acid	11.2	255*sh*	799.06359	0.27	C_34_H_24_O_23_	**799**, 497, 301								+				+		
4	Unknown	13.25	310	281.02976	−1.48	C_12_H_10_O_8_	**281**, 237	**283**, 191, 163			+				+	+	+				
5	*O*-Caffeoylglucaric acid isomer	15.26	298, 326	371.06060	−1.98	C_15_H_16_O_11_	**371**, 209, 191							+	+			+		+	[25]
6	Pedunculagin α or β	15.74	260*sh*	783.06839	−0.48	C_34_H_24_O_22_	**783**, 481, 301			+	+	+	+	+	+	+	+	+	+	+	(s)
7	Procyanidin B-type trimer	15.9	278	865.19739	−1.98	C_45_H_38_O_18_	**865**, 575, 289	**867**, 579, 291	+												
8	Bis-HHDP-glucose	16.58	260*sh*	783.06730	−1.29	C_34_H_24_O_22_	**783**, 481, 301									+	+				
9	*O*-Caffeoylglucaric acid isomer	17.75	310*sh*, 326	371.06162	−1.12	C_15_H_16_O_11_	**371**, 209, 191				+			+	+		+	+		+	[25]
10	5-*O*-caffeoylquinic acid	20.09	295*sh*, 325	353.08747	−0.96	C_16_H_18_O_9_	**353**, 191, 179	355, **163**										+	+		(s)
11	Catechin or epicatechin *O*-hexoside isomer	20.35	278	451.12458	−1.55	C_21_H_24_O_11_	**451**, 289, 245	**291**	+				+								
12	*O*-*p*-Coumaroylglucaric acid isomer	20.42	312	355.06630	−1.84	C_15_H_16_O_10_	**355**, 209, 191, 147										+	+		+	[25]
13	Dihydroxybenzoic acid *O*-pentoside	21.9	280	285.06146	−0.55	C_12_H_14_O_8_	**285**, 152											+		+	
14	Methylgallate *O*-glucoside	22.01	268	345.08239	−1.59	C_14_H_18_O_10_	**345**, 183, 168	**185**											+	+	
15	*O*-Caffeoylglucaric acid isomer	22.05	300, 326	371.06085	−3.3	C_15_H_16_O_11_	**371**, 209, 191							+	+		+	+			[25]
16	Galloyl-HHDP-glucose	22.26	250*sh*	633.07245	−0.24	C_27_H_22_O_18_	**633**, 301							+		+					
17	*O*-*p*-Coumaroylglucaric acid isomer	23.12	312	355.06611	−2.36	C_15_H_16_O_10_	**355**, 209, 191, 147										+	+		+	[25]
18	Pedunculagin α or β	23.3	260*sh*	783.06805	−0.82	C_34_H_24_O_22_	**783**, 481, 301	**303**		+	+	+	+	+	+	+	+	+	+	+	(s)
19	Procyanidin B-type dimer *O*-hexoside	23.36	280	739.18619	−2.06	C_36_H_36_O_17_	**739**, 451, 289	**741**, 579, 291	+												
20	Digalloyl-HHDP-gluconic acid	23.89	274	801.07970	1.57	C_34_H_26_O_23_	**801**, 633, 301, 169			+									+		
21	Galloyl-HHDP-glucose	24.13	280*sh*	633.07357	0.4	C_27_H_22_O_18_	**633**, 481, 301			+		+	+	+	+	+	+	+	+	+	
22	Catechin or epicatechin *O*-hexoside isomer	24.45	280	451.12352	−2.38	C_21_H_24_O_11_	**451**, 289	453, **291**	+					+							
23	Catechin or epicatechin *C*-hexoside isomer	25.22	280	451.12257	−4.09	C_21_H_24_O_11_	**451**, 289, 271	453, **291**	+												
24	Procyanidin A-type tetramer	25.6	280	1151.24372	−2.22	C_60_H_48_O_24_	**1151**, 863, 575, 289	**1153**, 865, 577, 291	+												
25	Procyanidin B1	25.9	280	577.13480	−0.55	C_30_H_26_O_12_	**577**, 289	**579**, 289, 257	+	+		+		+			+				(s)
26	*O*-Feruloylglucaric acid isomer	25.95	282, 326	385.07702	−2.26	C_16_H_18_O_11_	**385**, 209, 191, 147							+			+	+			[25]
27	Procyanidin B3	26.25	280	577.13426	−1.28	C_30_H_26_O_12_	**577**, 289	**579**, 289, 257				+	+	+				+	+	+	(s)
28	Catechin	27.05	280	289.07096	−2.33	C_15_H_14_O_6_	**289**, 245	**291**, 139	+	+	+	+	+	+			+	+	+	+	(s)
29	Caffeic acid	27.55	292, 320*sh*	179.03455	−2.54	C_9_H_8_O_4_	**179**, 135	**181**								+					(s)
30	*O*-Feruloylglucaric acid isomer	27.98	300*sh*, 318	385.07654	−2.63	C_16_H_18_O_11_	**385**, 209, 191, 147							+			+	+		+	[25]
31	Digalloyl-pentose	28.02	278	453.06751	0.08	C_19_H_18_O_13_	**453**, 301				+					+					
32	3-*O*-caffeoylquinic acid	28.35	295*sh*, 326	353.08691	−2.82	C_16_H_18_O_9_	**353**, 191	355, **163**					+					+		+	(s)
33	Digalloyl-HHDP-glucose	28.56	275	785.08401	−0.87	C_34_H_26_O_22_	**785**, 301, 275						+						+		
34	Procyanidin C2	28.87	280	865.19788	−0.51	C_45_H_38_O_18_	**865**, 575, 289	**867**, 579, 289	+			+	+	+						+	[23]
35	Digalloylglucose isomer	29.32	276	483.07714	−2.2	C_20_H_20_O_14_	**483**, 169, 125						+								
36	Digalloylglucose isomer	29.98	278	483.07745	−1.11	C_20_H_20_O_14_	**483**, 169, 125			+			+	+		+	+				
37	Laevigatin E isomer	30.2	274	1265.13990	1.26	C_54_H_42_O_36_	**1265**, 632, 301					+								+	
38	Methylgalloylmalic acid	30.77	278	299.04042	−1.91	C_12_H_12_O_9_	**299**, 183, 168, 133												+		
39	Quercetin *O*-hexoso-*O*-deoxyhexoso-hexoside	30.9	254, 354	771.19840	−0.79	C_33_H_40_O_21_	**771**, 609, 462, 299	**773**, 611, 465, 303		+											
40	Laevigatin E isomer	31.07	275*sh*	1265.13669	−1.27	C_54_H_42_O_36_	**1265**, 632, 301							+							
41	Catechin or epicatechin *O*-hexoside isomer	31.43	280	451.12343	−2.47	C_21_H_24_O_11_	**451**, 289	**453**, 291	+												
42	Procyanidin B-type trimer	31.54	280	865.19810	0.01	C_45_H_38_O_18_	**865**, 575, 289	**867**, 579, 291	+				+	+							
43	Galloyl-HHDP-glucose	31.73	272	633.07313	−0.04	C_27_H_22_O_18_	**633**, 481, 301			+	+		+	+	+	+	+	+	+	+	
44	Quercetin *O*-hexoso- *O*-hexoso-pentoside	31.82	256, 354	757.18267	−0.82	C_32_H_38_O_21_	**757**, 462, 299	**759**, 627, 465, 303		+											
45	Procyanidin B-type trimer	32.01	280	865.19703	−1.1	C_45_H_38_O_18_	**865**, 575, 289	**867**, 579, 291	+			+									
46	Brevifolincarboxylic acid	32.32	278, 360	291.01408	−1.94	C_13_H_8_O_8_	**291**, 247	**293**		+	+		+	+	+	+	+	+	+	+	
47	Procyanidin B2	33.71	278	577.13502	−0.04	C_30_H_26_O_12_	**577**, 289	**579**, 291, 139	+												(s)
48	Galloyl-HHDP-glucose	33.8	256, 342	633.07336	−0.48	C_27_H_22_O_18_	**633**, 481, 301										+			+	
49	Ellagic acid derivative	33.84	280*sh*	898.13120	2.12	C_36_H_35_O_27_	**898**, 783, 633, 301			+				+							
50	Brevifolincarboxylic acid isomer	34.01	284*sh*, 342	291.01448	−0.94	C_13_H_8_O_8_	**291**, 247								+					+	
51	Quercetin *O*-hexoso- *O*-hexoso-pentoside	34.03	254, 342	757.18241	−0.85	C_32_H_38_O_21_	**757**, 595, 462, 299	**759**, 597, 465, 303											+		
52	Laevigatin E isomer	34.19	275*sh*	1265.13618	−1.68	C_54_H_42_O_36_	**1265**, 632, 463, 301							+							
53	Quercetin *O*-hexoso- *O*-uronic acid derivative	34.2	254, 346	639.11995	−0.44	C_27_H_28_O_18_	**639**, 463, 300	**641**, 479, 303		+								+		+	
54	Procyanidin A-type tetramer	34.46	280	1151.24437	−1.65	C_60_H_48_O_24_	**1151**, 863, 575, 289	**1153**, 865, 577, 287	+												
55	Quercetin *O*-diuronic acid derivative	34.5	256, 352	653.09909	−0.32	C_27_H_26_O_19_	**653**, 447, 301	**655**, 479, 303			+		+	+				+			
56	Procyanidin B-type tetramer	34.54	280	1153.26132	−0.53	C_60_H_50_O_24_	**1153**, 576, 289	**1155**, 867, 577, 289				+									
57	*p*-Coumaroylquinic acid isomer	35.14	312	337.09286	0.02	C_16_H_18_O_8_	**337**, 191	**339**, 147										+			
58	Quercetin *O*-hexoso-*O*-hexoside	35.15	256, 352	625.14017	−1.24	C_27_H_30_O_17_	**625**, 462, 299	**627**, 465, 303		+						+	+		+		
59	Quercetin *O*-hexoso-*O*-uronic acid derivative	35.4	254, 346	639.12105	−2.16	C_27_H_28_O_18_	**639**, 463, 301	**641**, 465, 303		+				+		+	+			+	
60	Digalloyl-HHDP-glucose	35.47	276	785.08456	−0.06	C_34_H_26_O_22_	**785**, 615, 301, 169						+						+		
61	Epicatechin	35.74	280	289.07181	0.6	C_15_H_14_O_6_	**289**, 245	**291**, 139	+			+									(s)
62	Kaempferol *O*-hexoso-*O*-uronic acid derivative	36	264, 338	623.12581	0.57	C_27_H_28_O_17_	**623**, 284	**625**, 463, 287			+										
63	Quercetin *O*-hexoso- *O*-hexoso-deoxyhexoside	36.03	256, 348	771.19893	−0.45	C_33_H_40_O_21_	**771**, 462, 299	**773**, 627, 465, 303								+			+		
64	Procyanidin B-type trimer	36.07	280	865.19912	0.04	C_45_H_38_O_18_	**865**, 575, 289	**867**, 579, 291				+	+								
65	Caffeoylisocitric acid	36.32	300*sh*, 328	353.05058	−2.24	C_15_H_14_O_10_	**353**, 191, 179, 173, 155									+	+		+	+	[25]
66	Procyanidin B-type dimer	36.35	280	577.13479	−0.45	C_30_H_26_O_12_	**577**, 289	**579**, 291				+	+								
67	Kaempferol *O*-hexoso-deoxyhexoso-*O*-uronic acid derivative	37.05	266, 346	769.18282	−0.4	C_33_H_38_O_21_	**769**, 284	**771**, 625, 463, 287			+										
68	Dimeric hydrolysable tannin	37.55	270	1569.15737	−1.82	C_68_H_50_O_44_	**1569**, 784, 469, 301													+	
69	Valoneic acid dilactone	37.6	255*sh*, 362	469.00441	−0.32	C_21_H_10_O_13_	**469**, 425, 301								+		+				[26]
70	Ellagic acid derivative	37.76	268, 342	741.18713	−0.79	C_32_H_38_O_20_	**741**, 579, 446, 301												+		
71	Procyanidin A-type trimer	37.81	280	863.18110	−2.08	C_45_H_36_O_18_	**863**, 711, 573, 411, 289	**865**, 575, 287	+												
72	Kaempferol *O*-hexoso-deoxyhexoso-deoxyhexoso- *O*-uronic acid derivative	37.86	266, 346	915.24077	−0.13	C_39_H_48_O_25_	**915**, 285	**917**, 771, 625, 463, 287			+										
73	Galloyl-ellagic acid-*O*-hexoside	38.3	250, 374	615.06204	−1.43	C_27_H_20_O_17_	**615**, 463, 301					+		+			+	+		+	
74	Quercetin *O*-uronic acid derivative	38.4	254, 350	725.11985	−0.75	C_30_H_30_O_21_	**725**, 505, 300	**727**, 479, 303		+								+		+	
75	*O*-Caffeoylmalic acid	38.54	298, 326	295.04504	−2.45	C_13_H_12_O_8_	591, **295**, 179, **133**	**295**, **135**									+	+		+	[27]
76	Methylgalloyl-galloyl-glucose	38.85	270	497.09317	−1.17	C_21_H_22_O_14_	**497**, 345, 183, 169												+	+	
77	Sanguisorbic acid dilactone	38.88	255*sh*, 362	469.00439	0.38	C_21_H_10_O_13_	**469**, 425, 301								+						[26]
78	Kaempferol *O*-diuronic acid derivative	38.92	265, 350	637.10483	0.37	C_27_H_26_O_18_	**637**, 461, 285	**639**, 463, 287			+			+				+			
79	Quercetin *O*-(malonyl- hexoso)-*O*-hexoside	39.08	256, 354	711.14146	−0.09	C_30_H_32_O_20_	**711**, 667, 462, 299	**713**, 551, 465, 303		+				+					+		
80	Trigalloylglucose isomer	39.1	276	635.08854	−0.04	C_27_H_24_O_18_	635, 465, 313, **169**						+				+		+		[28]
81	Kaempferol *O*-hexoso- *O*-hexoside	39.31	262, 348	609.14565	−1.26	C_27_H_30_O_16_	**609**, 446, 283	**611**, 449, 287									+			+	
82	Ellagic acid *O*-uronic acid derivative	39.4	252, 362	477.03029	−1.24	C_20_H_14_O_14_	**477**, 301					+									[23]
83	Brevifolin	39.6	275, 350	247.02448	−1.92	C_12_H_8_O_6_	**247**, 191	**249**			+		+		+	+	+				[28]
84	Laevigatin isomer	39.77	255	1567.14302	−0.99	C_68_H_48_O_44_	1567, **783**, 633, 301												+	+	[23]
85	Quercetin *O*-deoxyhexoso- *O*-deoxyhexoso-hexoside	39.8	256, 354	755.20392	−0.22	C_33_H_40_O_20_	**755**, 609, 446, 299	**757**, 611, 449, 303		+											
86	Trigalloylglucose isomer	40.05	276	635.08863	−0.5	C_27_H_24_O_18_	635, 465, 313, **169**						+								[28]
87	Isorhamnetin *O*-hexoso- *O*-hexoso-pentoside	40.1	254, 352	771.19887	−0.64	C_33_H_40_O_21_	**771**, 476, 315, 300	**773**, 641, 479, 317											+		
88	Procyanidin B-type dimer *O*-gallate	40.28	278	729.14551	−0.59	C_37_H_30_O_16_	**729**, 577, 559, 289, 169	**731**, 289					+								
89	Syringic acid derivative	40.44	280	313.05569	−2.4	C_13_H_14_O_9_	**313**, 197, 182												+		
90	Procyanidin A-type trimer	40.52	280	863.18180	−0.75	C_45_H_36_O_18_	**863**, 573, 411, 289	**865**, 287	+												
91	Isorhamnetin *O*-hexoso- *O*-uronic acid derivative	40.67	266, 348	653.13595	0.25	C_28_H_30_O_18_	**653**, 477, 314	**655**, 479, 317												+	
92	Apigenin *C*-dihexoside	40.75	270, 332	593.15106	−0.09	C_27_H_30_O_15_	**593**, 473, 383, 353	**595**, 439, 355, 325		+								+		+	
93	Procyanidin B-type tetramer	40.8	280	1153.25158	−0.3	C_60_H_50_O_24_	**1153**, 863, 576, 289	**1155**, 865, 577, 289				+									
94	Procyanidin C1	41.2	280	865.19784	−0.81	C_45_H_38_O_18_	**865**, 577, 289	**867**, 579, 291	+												(s)
95	Ellagic acid *O*-uronic acid derivative	41.47	252, 360	477.03021	−0.63	C_20_H_14_O_14_	**477**, 301						+								
96	Procyanidin A-type tetramer	41.51	280	1151.24448	−1.56	C_60_H_48_O_24_	**1151**, 863, 575, 289	**1153**, 865, 577, 287	+												
97	Ellagic acid *O*-hexoside	41.64	252, 362	463.05127	−0.58	C_20_H_16_O_13_	**463**, 301					+	+		+	+	+	+	+	+	[23]
98	Isorhamnetin *O*-diuronic acid derivative	41.8	254, 352	667.11526	0.2	C_28_H_28_O_19_	1335, **667**, 491, 315	**669**, 493, 317			+			+				+		+	
99	Quercetin *O*-deoxyhexoso- *O*-hexoso-pentoside	42.35	254, 352	741.18832	−0.66	C_32_H_38_O_20_	**741**, 446, 299	**743**, 611, 449, 303		+											
100	Isorhamnetin *O*-hexoso- *O*-uronic acid derivative	42.43	254, 352	653.13560	−0.3	C_28_H_30_O_18_	**653**, 477, 315	**655**, 479, 317								+					
101	Ellagic acid *O*-hexoside	42.7	250, 370	463.05107	−1.61	C_20_H_16_O_13_	**463**, 301					+		+			+		+		[23]
102	Agrimonic acid A or B	43.11	270*sh*	1103.08618	0.64	C_43_H_32_O_31_	**1103**, 935, 783, 301, 169					+						+		+	[23]
103	Tetragalloylglucose isomer	43.13	278	787.10004	−0.98	C_34_H_28_O_22_	**787**, 465, 169						+								[28]
104	Kaempferol *O*-deoxyhexoso-hexoso-*O*-deoxyhexoside	43.16	266, 346	739.20805	−0.74	C_33_H_40_O_19_	**739**, 593, 430, 283	**741**, 595, 433, 287		+										+	
105	Quercetin *O*-hexoso-*O*-hexoside	43.17	264, 344	625.14038	−1.14	C_27_H_30_O_17_	**625**, 463, 300	**627**, 465, 303									+				
106	Catechin or epicatechin *O*-hexoside isomer	43.3	278	451.12511	0.25	C_21_H_24_O_11_	**451**, 289	**289**					+								
107	Procyanidin B-type dimer	43.6	280	577.13537	−0.24	C_30_H_26_O_12_	**577**, 289	**579**, 287	+				+								
108	Galloyl-bis-HHDP-glucose	44.2	255	935.08057	−0.13	C_41_H_28_O_26_	**935**, 633, 467, 301			+	+		+		+	+	+		+	+	[20]
109	Laevigatin isomer	44.6	255	1567.14331	−0.8	C_68_H_48_O_44_	1567, **783**, 633, 301			+	+	+	+	+				+	+	+	[23]
110	Procyanidin A-type tetramer	45.03	280	1151.24657	0.26	C_60_H_48_O_24_	**1151**, 863, 575, 289	**1153**, 865, 577, 289	+												
111	Quercetin *O*-hexoso-*O*-hexoside	45.39	254, 346	625.14019	−1.3	C_27_H_30_O_17_	**625**, 463, 300	**627**, 465, 303									+				
112	HHDP-NHTP-glucose	45.56	254	933.06390	−0.43	C_41_H_26_O_26_	**933**, 631, 466, 301				+				+	+					
113	Kaempferol *O*-diuronic acid derivative	45.86	266, 336	637.10484	−0.55	C_27_H_26_O_18_	**637**, 461, 285	**639**, 463, 287		+											
114	Laevigatin isomer	45.99	255	1567.14487	0.19	C_68_H_48_O_44_	1567, **783**, 301							+				+			[23]
115	Procyanidin B-type pentamer	46.13	278	1441.32708	1.23	C_75_H_62_O_30_	**1441**, 1153, 863, 575, 289	**1443**, 1155, 865, 577, 289				+									
116	Quercetin *O*-hexoso-*O*-uronic acid derivative	46.5	264, 340*sh*	639.12041	0.17	C_27_H_28_O_18_	**639**, 463, 301	**641**, 465, 303									+				
117	Kaempferol *O*-hexoso- *O*-hexoso-deoxyhexoside	46.63	264, 350*sh*	755.20300	−0.67	C_33_H_40_O_20_	**755**, 593, 447, 285	**757**, 595, 449, 287								+		+		+	
118	Galloyl-bis-HHDP-glucose	47.7	275*sh*	935.07950	−0.36	C_41_H_28_O_26_	**935**, 633, 467, 301								+	+	+			+	
119	Apigenin *C*-hexoso-*C*-pentoside	49.2	268, 340	563.13970	−1.45	C_26_H_28_O_14_	**563**, 519, 473, 443, 383, 353	**565**, 379, 355, 325												+	
120	Quercetin *O*-hexoso-pentoside	50.08	255, 352	595.13009	0.61	C_26_H_28_O_16_	**595**, 300	597, 465, **303**					+		+		+				
121	Quercetin *O*-hexoso-pentoside	50.95	256, 354	595.12982	−0.59	C_26_H_28_O_16_	**595**, 300	**597**, 465, 303		+			+				+				
122	Quercetin *O*-deoxyhexoso- *O*-uronic acid derivative	51.17	254, 352	623.12438	−1.13	C_27_H_28_O_17_	**623**, 301	**625**, 479, 303										+			
123	Feruloylisocitric acid	51.46	284, 326	367.06617	−2.08	C_16_H_16_O_10_	**367**, 173									+	+			+	[25]
124	Laevigatin isomer	51.56	255	1567.14348	−0.69	C_68_H_48_O_44_	1567, **783**, 301			+		+	+	+				+		+	[23]
125	Isorhamnetin *O*-hexoso-hexoside	51.65	270, 350	639.15591	−1.2	C_28_H_32_O_17_	**639**, 314, 300	641, 479, **317**						+	+	+	+	+	+	+	
126	Quercetin *O*-deoxyhexoso-hexoside	52.15	256, 354	609.11075	−0.28	C_27_H_30_O_16_	**609**, 300	**611**, 449, 303		+											
127	Quercetin *O*-galloyl-hexose	52.7	264, 352	615.09845	−1.15	C_28_H_24_O_16_	**615**, 463, 300, 169	**617**, 303					+								
128	Laevigatin isomer	52.76	255	1567.14432	−0.15	C_68_H_48_O_44_	1567, **783**, 301					+		+			+	+		+	[23]
129	Quercetin *O*-pentoso-hexoside	53.41	256, 354	595.12983	−1.16	C_26_H_28_O_16_	**595**, 300	**597**, 435, 303											+		
130	Ellagic acid *O*-methyl ether *O*-uronic acid derivative	54.1	254, 360	491.04680	0.22	C_21_H_16_O_14_	**491**, 315, 299.9					+			+		+	+		+	[23]
131	Trigalloyl-HHDP-glucose	54.3	270	937.09372	−1.14	C_41_H_30_O_26_	**937**, 783, 468, 301, 169			+				+					+		
132	Ellagic acid *O*-pentoside	56.35	252, 360	433.04120	−0.54	C_19_H_14_O_12_	**463**, 301					+			+	+					[23]
133	Trigalloyl-HHDP-glucose	56.57	278	937.09504	−0.77	C_41_H_30_O_26_	**937**, 468, 301												+		
134	Quercetin *O*-galloyl-hexose	57.03	264, 354	615.09882	−1.5	C_28_H_24_O_16_	**615**, 463, 300, 169	**617**, 303					+								
135	Ellagic acid	57.7	254, 370	300.99864	−0.98	C_14_H_6_O_8_	**301**, 275	**303**		+	+	+	+	+	+	+	+	+	+	+	(s)
136	Isoquercitrin (Quercetin 3-*O*-glucoside)	59.8	254, 354	463.08790	−0.94	C_21_H_20_O_12_	**463**, 300, 271	**465**, 303					+		+		+	+		+	(s)
137	Tetragalloylglucose isomer	62	278	787.09906	−1.22	C_34_H_28_O_22_	**787**, 465, 169						+							+	[28]
138	Rutin (Quercetin 3-*O*-rutinoside)	63.7	256, 354	609.14556	−0.6	C_27_H_30_O_16_	**609**, 300, 271	**611**, 465, 303					+			+		+	+		(s)
139	Hyperoside (Quercetin 3-*O*-galactoside)	64.13	255, 355	463.08829	−0.76	C_21_H_20_O_12_	**463**, 300	**465**, 303		+			+	+		+	+	+	+	+	(s)
140	Kaempferol *O*-hexoso-pentoside	65.75	266, 348	579.13529	−0.02	C_26_H_28_O_15_	**579**, 284	581, 449, **287**							+						
141	Quercetin *O*-uronic acid derivative	66.03	256, 354	477.06730	−0.23	C_21_H_18_O_13_	**477**, 301	**479**, 303			+		+	+		+	+	+		+	
142	Kaempferol *O*-hexoside	67.4	252, 350	447.09290	−1.32	C_21_H_20_O_11_	**447**, 284	**449**, 287		+								+			
143	Kaempferol *O*-uronic acid derivative	68.88	254, 348	461.07221	−1.04	C_21_H_18_O_12_	**461**, 285	**463**, 287		+											
144	Galloyl-bis-HHDP-glucose	69.66	260*sh*	935.07940	−0.31	C_41_H_28_O_26_	**935**, 467, 301			+		+	+	+	+	+	+	+	+	+	
145	Quercetin *O*-pentoso-*O*- pentoso-uronic acid derivative	71.48	254, 352	739.17255	−0.32	C_32_H_36_O_20_	**739**, 300	**741**, 609, 433, 303							+						
146	Dimeric ellagitannin	72.47	270	1871.16610	−0.21	C_82_H_56_O_52_	1871, 1265, **935**, 783, 301												+		
147	Kaempferol *O*-deoxyhexoso-hexoside	72.86	266, 345*sh*	579.13479	−1.17	C_26_H_28_O_15_	**579**, 284	**581**, 449, 287											+		
148	Kaempferol *O*-deoxyhexoso-deoxyhexoso-*O*-hexoside	74.17	264, 346	739.20843	−0.33	C_33_H_40_O_19_	**739**, 593, 284	**741**, 595, 449, 287			+										
149	Quercetin 3-*O*-arabinoside	85.55	254*sh*, 350	433.07623	−3.28	C_20_H_18_O_11_	**433**, 300	435, **303**					+								(s)
150	Isorhamnetin *O*-pentoso-hexoside	86.15	254, 354	609.14566	−1.28	C_27_H_30_O_16_	**609**, 314, 300	**611**, 479, 317											+		
151	Dimeric ellagitannin	87.1	250*sh*	1869.14746	−1.81	C_82_H_54_O_52_	1869, **934**, 783, 301												+	+	
152	Quercetin *O*-pentoso-deoxyhexoside	87.15	256, 352	579.13374	−3.01	C_26_H_28_O_15_	**579**, 300	**581**, 435, 303					+								
153	Kaempferol *O*-deoxyhexoso-*O*-hexoside	88.6	264, 346	593.15030	−1.35	C_27_H_30_O_15_	**593**, 447, 284	**595**, 449, 287								+		+	+		
154	Quercetin derivative	88.75	256, 350	607.12966	−1.56	C_27_H_28_O_16_	**607**, 300	**609**, 303												+	
155	Astragalin (Kaempferol 3-*O*-glucoside)	88.8	264, 350	447.09299	−0.34	C_21_H_20_O_11_	**447**, 284	**449**, 287		+	+		+	+	+	+	+	+	+	+	(s)
156	Quercetin *O*-deoxyhexoso-hexoside	89.1	256, 348	609.14510	−1.89	C_27_H_30_O_16_	**609**, 300	**611**, 448, 303					+								
157	Kaempferol 3-*O*-glucuronide	89.25	265, 350	461.07183	−1.38	C_21_H_18_O_12_	**461**, 285	**463**, 287						+		+	+				(s)
158	Isorhamnetin *O*-deoxyhexoso-deoxyhexoso-*O*-hexoside	89.33	254, 352	769.21962	−0.11	C_34_H_42_O_20_	**769**, 315	**771**, 625, 479, 317			+										
159	Kaempferol *O*-uronic acid derivative	89.6	268, 342*sh*	461.07133	−2.38	C_21_H_18_O_12_	**461**, 285	**463**, 287					+				+				
160	Kaempferol *O*-deoxyhexoso- *O*-hexoso-deoxyhexoside	89.7	266, 348	737.19349	0.28	C_33_H_38_O_19_	**737**, 593, 284	**739**, 593, 433, 287												+	
161	Apigenin *O*-hexoside	90.1	266, 336	431.09795	−1.23	C_21_H_20_O_10_	**431**, 268	**433**, 271										+			
162	Agrimoniin	90.3	250*sh*	1869.14917	−0.89	C_82_H_54_O_52_	1870, 1085, **934**, 783, 301			+	+	+	+	+	+	+	+	+	+	+	(s)
163	Ellagic acid *O*-methyl ether *O*-pentoside	90.45	280*sh*, 365	447.05603	−1.61	C_20_H_16_O_12_	**447**, 315, 301								+						[28]
164	Kaempferol derivative	90.61	264, 348	723.17755	0.1	C_32_H_36_O_19_	**723**, 621, 579, 284	**725**, 593, 287							+						
165	Kaempferol derivative	91.08	266, 348	635.16048	−2.02	C_29_H_32_O_16_	**635**, 284	**637**, 287												+	
166	Apigenin *O*-uronic acid derivative	91.35	266, 336	445.07769	0.26	C_21_H_18_O_11_	891, **445**, 269	**447**, 271		+											
167	Isorhamnetin *O*-deoxyhexoso-hexoside	91.43	254, 352	623.16185	0.37	C_28_H_32_O_16_	**623**, 314	**625**, 479, 317								+			+		
168	Pentagalloylglucose isomer	91.7	280	939.11106	−0.58	C_41_H_32_O_26_	**939**, 769, 469, 169						+						+		
169	Isorhamnetin 3-*O*-glucoside	91.72	254, 350	477.10350	−0.23	C_22_H_22_O_12_	**477**, 314, 300	**479**, 317								+					(s)
170	Kaempferol derivative	91.94	266, 350	737.19229	−1.6	C_33_H_38_O_19_	**737**, 284	**739**, 593, 287												+	
171	Isorhamnetin *O*-deoxyhexoso-hexoso *O*-pentoside	92.02	254*sh*, 355	753.18765	−0.65	C_33_H_38_O_20_	**753**, 314	755, 623, **317**							+						
172	Isorhamnetin *O*-uronic acid derivative	92.79	254, 354	491.08194	−1.69	C_22_H_20_O_13_	**491**, 315, 300	**493**, 317						+		+		+			
173	Isorhamnetin *O*-pentoso- O-deoxyhexoso-*O*-uronic acid derivative	92.93	254, 354	767.20286	−1.49	C_34_H_40_O_20_	**767**, 621, 314	**769**, 623, 493, 317												+	
174	Chrysoeriol *O*-uronic acid derivative	93.7	266*sh*, 346	475.08764	−0.91	C_22_H_20_O_12_	951, **475**, 299	**477**, 301		+											
175	Kaempferol *O*-acetylhexoside	94.62	264, 346	489.10342	−1.01	C_23_H_22_O_12_	**489**, 284	**491**, 287									+	+	+	+	
176	Kaempferol derivative	94.8	266, 348	591.13497	−0.5	C_27_H_28_O_15_	**591**, 284	**593**, 287												+	
177	Quercetin *O*-uronic acid derivative	94.87	266*sh*, 360	477.06702	−0.36	C_21_H_18_O_13_	**477**, 301	**479**, 303		+	+			+				+			
178	Kaempferol *O*-malonylhexoside	94.97	266*sh*, 348	533.09266	−1.62	C_24_H_22_O_14_	**533**, 284	**535**, 287										+			
179	Isorhamnetin *O*-galloyldeoxyhexoside	95.2	270, 348	629.11322	−2.41	C_29_H_26_O_16_	**629**, 314, 299, 169	**631**, 317					+								
180	Naringenin O-hexoside	95.52	276*sh*, 362	433.11345	−1.15	C_21_H_22_O_10_	**433**, 271	435, **273**									+				
181	Isorhamentin derivative	95.83	254, 352	621.14502	−1.78	C_28_H_30_O_16_	**621**, 314, 300	**623**, 317												+	
182	Isorhamnetin *O*-acetylhexoside	95.96	254, 352	519.11432	−0.23	C_24_H_22_O_13_	**519**, 314, 299	**521**, 317								+			+	+	
183	Acacetin	96.26	254	283.06188	0.6	C_16_H_12_O_5_	**283**, 268	**285**, 242							+						[29]
184	Apigenin *O*-acetylhexoside	97.3	266, 326	473.10900	−0.77	C_23_H_22_O_11_	**473**, 269											+			
185	Apigenin	98.1	268, 338	269.04538	−1.93	C_15_H_10_O_5_	**269**	**271**										+			(s)
186	Quercetin *O*-deoxyhexoso- deoxyhexoso-*O*-hexoside	98.43	266, 346	753.22397	−0.86	C_34_H_42_O_19_	**753**, 299	**755**, 609, 463, 301			+										
187	Isorhamnetin *O*-hexoside	98.9	256, 356	477.10244	−3.05	C_22_H_22_O_12_	**477**, 314, 299	**479**, 317					+								
188	Isorhamnetin *O*-hexoside	99.21	256, 354	477.10227	−2.92	C_22_H_22_O_12_	**477**, 314, 271	**479**, 317					+								
189	Tricin *O*-deoxyhexoso- deoxyhexoso-*O*-hexoside	99.5	254, 354	783.23498	−0.36	C_35_H_44_O_20_	**783**, 329	**785**, 639, 493, 331			+										
190	*trans*-Tiliroside	101.41	268, 315	593.13011	0.27	C_30_H_26_O_13_	**593**, 284	**595**, 287		+	+		+	+	+	+	+	+	+	+	(s)
191	Isorhamnetin *O*-pentoside	101.5	258, 354	447.09386	−1.61	C_21_H_20_O_11_	**447**, 315, 271	**449**, 317					+								
192	Kaempferol derivative	101.87	268, 330	623.13981	−0.98	C_31_H_28_O_14_	**623**, 284	**625**, 287		+	+			+	+		+	+	+	+	
193	*cis*-Tiliroside	102.37	268, 315	593.12995	−0.3	C_30_H_26_O_13_	**593**, 284	**595**, 287		+	+		+	+	+	+	+	+	+	+	
194	Isorhamnetin *O*-deoxyhexoside	102.65	256, 350	461.10762	−2.95	C_22_H_22_O_11_	**461**, 314, 271	**463**, 317, 274					+								
195	Tricin *O*-uronic acid derivative	103.04	254*sh*, 352	505.09843	−0.58	C_23_H_22_O_13_	**505**, 329	**507**, 331, 316			+										
196	Isorhamnetin derivative	103.4	256, 350	593.14977	−2.29	C_27_H_30_O_15_	**593**, 314, 299	**595**, 317					+								
197	N^1^, N^5^, N^10^-Tricoumaroyl spermidine	104.47	295, 310*sh*	582.26028	−0.48	C_34_H_37_N_3_O_6_	**582**, 462, 342, 285	**584**, 438, 292, 147		+	+			+	+	+	+	+	+	+	[30]
198	Ellagic acid derivative	111.73	350*sh*, 362	422.99970	0.41	C_20_H_8_O_11_	**423**, 343, 269							+							

^1^ Exact mass of [M-H]^−^ ion; *sh*—peak shoulder; bold—most aboundantion; (s)—reference substance; HHDP—hexahydroxydiphenoyl group; NHTP—nonahydroxytriphenoyl group.

**Table 3 ijms-24-04836-t003:** IC_50_ values (concentration causing viability/proliferation inhibition by 50% compared to control) of aqueous acetone extracts isolated from selected *Potentilla* L. species. IC_50_ values were calculated for human colon epithelial cell line CCD841 CoN and human colon adenocarcinoma cell line LS180 based on results of MTT as well as BrdU assays performed after 48 h of cells’ treatment with investigated compounds.

Samples	MTT Assay						BrdU Assay					
	LS180			CCD841 CoN			LS180			CCD841 CoN		
	IC_50_ (µg/mL)	Trust Range (µg/mL)	R^2^	IC_50_ (µg/mL)	Trust Range (µg/mL)	R^2^	IC_50_ (µg/mL)	Trust Range (µg/mL)	R^2^	IC_50_ (µg/mL)	Trust Range (µg/mL)	R^2^
**PAL7r**	82	77–87	0.980	496	396–623	0.908	52	41–64	0.917	412	351–483	0.841
**PAU7**	192	180–206	0.920	1575	536–4632	0.672	1495	1311–1704	0.871	2058	1626–2604	0.542
**PER7**	176	166–186	0.957	672	474–952	0.891	1001	847–1183	0.845	3705	2368–5796	0.336
**PER7r**	110	101–120	0.943	523	326–839	0.595	54	44–66	0.925	337	281–405	0.856
**PFR7**	89	85–92	0.989	707	450–1113	0.737	50	40–62	0.916	282	244–327	0.809
**PHY7**	156	146–167	0.952	489	334–717	0.838	425	350–516	0.843	631	495–804	0.765
**PME7**	128	122–133	0.983	380	291–495	0.870	417	325–536	0.774	837	661–1061	0.740
**PNE7**	112	106–118	0.977	1795	329–9800	0.365	343	298–395	0.915	586	451–763	0.748
**PPE7**	158	150–167	0.966	620	367–1047	0.663	343	283–414	0.850	911	728–1140	0.768
**PPU7**	197	185–210	0.967	865	359–2081	0.531	881	761–1019	0.836	937	803–1093	0.846
**PRI7**	213	200–228	0.968	2402	788–7326	0.717	542	452–649	0.848	1230	837–1806	0.553
**PTH7**	225	215–236	0.956	969	443–2119	0.643	606	521–704	0.876	1039	791–1364	0.693
**5-FU**	31	28–33	0.977	113	81–157	0.884	15	13–16	0.956	94	80–111	0.933

**Table 4 ijms-24-04836-t004:** List of plants from the *Potentilla* genus that were screened in the study and extraction yields.

Sample Name	Lant Species	Voucher Specimen No.	Parts Used ^1^	Extraction Yield (%) ^2^
**PAL7r**	*Potentilla alba* L.	PAL-17039	R	11.2%
**PAU7**	*Potentilla aurea* L.	PAU-20045	A	32.8%
**PER7**	*Potentilla erecta* (L.) Raeusch	PER-06016	A	17.8%
**PER7r**			R	15.7%
**PFR7**	*Potentilla fruticosa* L. (syn. *Dasiphora fruticosa* (L.) Rydb.)	PFR-06018	L	36.6%
**PHY7**	*Potentilla hyparctica* Malte	PHY-20046	A	26.5%
**PME7**	*Potentilla megalantha* Takeda	PME-18043	A	34.1%
**PNE7**	*Potentilla nepalensis* Hook.	PNE-06023	A	33.4%
**PPE7**	*Potentilla pensylvanica* L.	PPS-08025	A	22.4%
**PPU7**	*Potentilla pulcherrima* Lehm.	PPU-18044	A	28%
**PRI7**	*Potentilla rigoi* Th. Wolf	PRI-20047	A	30.6%
**PTH7**	*Potentilla thuringiaca* Bernh.	PTH-06022	A	22.8%

^1^ A, aerial parts; L, leaves; R, rhizomes; ^2^ Extraction yield of purified fraction.

## Data Availability

The datasets used and/or analysed during the current study are available from the corresponding author upon reasonable request.

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
