# Peer review of "Phytochemical Profiling of Extracts from Rare Potentilla Species and Evaluation of Their Anticancer Potential"

_ijms, 2023, doi:10.3390/ijms24054836_

Round 1

Reviewer 1 Report

At first, i would like to thank  authors about their effort in preparing this scientific paper.

There are few comments

1. What is the indication of number 7 in PFR7, PHY7, PME7, ....?

2. What is LDH stands for?? Please, clarify it in abstract and in its first mention in manuscript

3. What is the difference between MTT assay and BrdU assay? Please, discuss the differences between them and why did you do both assays??

Author Response

Point-by-point response to specific comments:

Reviewer #1: At first, I would like to thank  authors about their effort in preparing this scientific paper. There are few comments

  1. What is the indication of number 7 in PFR7, PHY7, PME7, ....?
    Response: We appreciate your suggestion. The criteria for numbering extracts with the number 7 to unambiguously distinguish acetone extracts from extracts and fractions obtained with other solvents in our department.
  2. What is LDH stands for?? Please, clarify it in abstract and in its first mention in manuscript
    Response: We appreciate your suggestion. LDH stands for lactate dehydrogenase (LDH) release assay. The abstract and manuscript was supplemented with proper information.

  3. What is the difference between MTT assay and BrdU assay? Please, discuss the differences between them and why did you do both assays??
    Response: The MTT assay is a well-known and widely used assay for the evaluation of cell viability and/or proliferation, which is dependent on experiment conditions especially on cell density at the beginning of the experiment and the time of treatment. In the revised manuscript, the MTT assay was used for the examination of cell viability, which was clearly indicated. In metabolically active cells mitochondrial dehydrogenase reduces yellow, water-soluble yellow tetrazolium salt (MTT; 3-bromide (4,5-dimethyl-2-thiazolyl)-2,5-diphenyltetrazole) to blue formazan, which in form of water-insoluble crystals accumulates in the cells. The insoluble formazan crystals are dissolved using a solubilization solution (SDS-HCl buffer of pH 7.4) and the resulting colored solution is quantified by measuring absorbance at 500-600 nanometers using a spectrophotometer. The intensity of the color is directly proportional to the number of  viable, metabolically active cells. Despite the fact that MTT assay may be used for examination of cell proliferation, in the revised manuscript this phenomenon was investigated using a more specific and more sensitive BrdU immunoassay, which was also clearly indicated. BrdU (5-bromo-2'-deoxyuridine) is a synthetic thymidine nucleoside analog incorporated into newly synthesized DNA strands of actively proliferating cells. Following partial denaturation of double-stranded DNA, incorporated BrdU is detected by specific antibodies conjugated with an enzyme that catalyzes the reaction that results in the formation of a color product. The intensity of the color is directly proportional to the amount of living, dividing cells.

Reviewer 2 Report

The English of the conclusion and discussion sections, especially the introduction section of the manuscript, should be rearranged. Attention should be paid to the separation of words at the end of the line with '-'.

Author Response

Reviewer #2: The English of the conclusion and discussion sections, especially the introduction section of the manuscript, should be rearranged. Attention should be paid to the separation of words at the end of the line with '-'.

Response: The authors acknowledge the Reviewer’s suggestion, however, the authors confirmed that manuscript was checked by a native speaker before submission to the International Journal of Molecular Sciences (language editing MDPI services – English editing ID: 60291).

Reviewer 3 Report

In this manuscript, Augustynowicz et al., investigated the phytochemical and biological profiles of acetone extracts from selected Potentilla species. I think the data presented in this manuscript is interesting in the field of colon cancer research. However, I think the manuscript can be considered for publication after Major revisions:

1.      In the title it is better to avoid abbreviations.

2.      I think the work is incomplete and the authors should carry out additional experiments to explore the mechanism by which the promising extract/compounds induce the anti-proliferation activity such as cell cycle analysis, apoptotic effects and activation of caspase proteases. To claim the anticancer activity, the authors need to investigate whether this occurred or not otherwise it can be considered as a cytotoxic study (Not anticancer study as in the title).

3.      From the IC50 values in in Tables 3, it can be concluded that "No statistical analysis was performed (No standard deviation). Did the authors repeat the experiments?

4.      Single cancer cell line is not recommended. Authors need to strongly justify why only 1 cancer cell line was selected and why only this particular cell line was selected.

5.      In the discussion part, authors need to justify the high difference in the IC50 values (between MTT assay and BrdU assay) as well as to compare the obtained results with previous studies.

6.      Authors need to provide the voucher specimens number.

7.      What was the solvent that used to dissolve the acetone extract before treating the cells (the vehicle)?

8.      The conclusion paragraph looks like a repetition of the results. Authors should indicate what is new and innovative in the investigation, gaps in this investigation and how to close them as well as indicating the holistic conclusion and recommendations.

Author Response

Reviewer #3: In this manuscript, Augustynowicz et al., investigated the phytochemical and biological profiles of acetone extracts from selected Potentilla species. I think the data presented in this manuscript is interesting in the field of colon cancer research. However, I think the manuscript can be considered for publication after Major revisions:

  1. In the title it is better to avoid abbreviations.

    Response: We appreciate your suggestion. Therefore the title of the manuscript has been changed.

  2. I think the work is incomplete and the authors should carry out additional experiments to explore the mechanism by which the promising extract/compounds induce the anti-proliferation activity such as cell cycle analysis, apoptotic effects and activation of caspase proteases. To claim the anticancer activity, the authors need to investigate whether this occurred or not otherwise it can be considered as a cytotoxic study (Not anticancer study as in the title).

    Response: We acknowledge the Reviewer’s comment that other studies are required to verify discovered beneficial effect of tested extracts and admit that such studies will be conducted in the future in order to better understand the molecular mechanism of the most promising extracts. Nevertheless, we cannot agree that without the results of cell cycle analysis or examination of proapoptotic properties, the manuscript is incomplete. We would to emphasize that the presented work is the preliminary studies conducted on both cancer and normal cells, which allows for investigating the selectivity of tested extracts, which is not a very common procedure in this type of study. Furthermore, in spite of the Reviewer's suggestion presented in vitro studies have been designed in the way to investigate some mechanisms of investigated extracts' action – LDH assay allowed to check the impact of the compounds on the cell membranes integrity, MTT assay allowed to examine the influence of the extracts on cell metabolic activity, while BrdU assay allowed to evaluate the compounds antiproliferative properties by monitoring of BrdU incorporation to newly synthesized DNA in dividing cells. Because of that, we cannot agree that the paper is just a cytotoxic study. Furthermore, in light of the Reviewer's comment, which if we understand correctly suggests that the manuscript creates the impression that we have discovered a cure for cancer, we would like to stress that we have tried to choose words and phrases very carefully so as not to give the above-mentioned wrong and unintentional impression. That is why we deliberately used the term anticancer potential in the title, not the anticancer effect or action. Nevertheless, in light of the comment of the reviewer, we decided to supplement the Conclusions and redraft the summary of the abstract in such a way that clearly presents the importance of our studies and their position in the drug/diet supplement development pathway. All made changes have been highlighted in red font.

  3. From the IC50 values in Tables 3, it can be concluded that "No statistical analysis was performed (No standard deviation). Did the authors repeat the experiments?

    Response: IC50 was determined using Graph Pad Prism based on the results of MTT and BrdU tests for individual cell lines. The same numerical data as presented in Figures 1 and 2 were used to determine the IC50 value of the tested extracts. However, to determine the IC50 value of fluorouracil, the results of MTT and BrdU assays were carried out in cells treated with the cytostatic at the concentrations of 5, 10, 25 and 50 µM. According to Reviewer suggestion the Table 3 has been supplemented with confidence intervals and R2.

  4. Single cancer cell line is not recommended. Authors need to strongly justify why only 1 cancer cell line was selected and why only this particular cell line was selected.

    Response: Due to the fact that our previous study on the chemopreventive potential of acetone extracts from selected Potentilla species was carried out on human colon adenocarcinoma cell line LS180 and human colonic epithelial cell line CCD841 CoN, we decided to use the same research model for the continuation of the investigation. Both cell lines were carefully selected based on a review of literature data dedicated to the analysis of extracts’ influence on colon cancer cells. Both cell lines were obtained from well-known the European Collection of Cell Cultures. Both cell lines were derived from humans: CCD 841 CoN cells were isolated from normal human colon tissue (it needs to be highlighted that CCD841 CoN unlike the CCD 841 CoTr cell, is not immortalized by the virus SV40); LS180 cells were isolated from the colon of patient with colon adenocarcinoma on stage B according to Dukes' classification. Nevertheless, we acknowledge the Reviewer’s suggestion to carry out analyses on a larger number of cell lines and we are going to extend the panel of investigated cancer cells in the future. Nonetheless, we would like to highlight that the presented study was conducted in both cancer and normal cell lines, and all obtained data was presented in Figures 1-3 as well as Table 3. We also would like to emphasize that the presented work is one of the few studies in which the assessment of the chemopreventive potential of a substance is carried out simultaneously on normal and cancer cells under the same conditions and using the same research methods. The presented approach, which is unusual (most studies focused only on cancer cells) allows for determining the selectivity and safety of the use of potential active substances at the in vitro level.

  5. In the discussion part, authors need to justify the high difference in the IC50 values (between MTT assay and BrdU assay) as well as to compare the obtained results with previous studies.

    Response: The IC50 value is dependent on the assay conditions. In the case of in vitro study, the IC50 value is calculated for the same compounds at the same concentrations (conditions of an experiment as well as the type of assay used for the study), but in different models (different cell lines), or after different time of treatment, will be different. Data presented in Table 3 perfectly illustrate these relationships. Although the analysis was performed under the same conditions (same cell line, same cell density, same exposure time, same tested concentrations), using different assays resulted in different results. Therefore, the presentation of IC50 values requires the specification of the above-mentioned experiment conditions, which in our opinion was done in the manuscript. Moreover, the discussion part already includes a comparison of obtained results with our previous studies.

  6. Authors need to provide the voucher specimens number.

    Response: Thank you for this valuable suggestion. Therefore, in materials and methods section the Table 4 was added with proper information.

  7. What was the solvent that used to dissolve the acetone extract before treating the cells (the vehicle)?

    Response: We acknowledge the Reviewer’s comment, nevertheless the issue raised by the Reviewer is described in the manuscript in the section 3.1. Chemicals as follows “Investigated extracts (100 mg/mL) and 5-fluorouracil (50 mM) were dissolved in dimethyl sulfoxide (DMSO) to prepare stock solutions. Working solutions were prepared by dissolving stock solutions in a culture medium. The final concentration of DMSO in all working solutions used in the studies was 0.25%.”

  8. The conclusion paragraph looks like a repetition of the results. Authors should indicate what is new and innovative in the investigation, gaps in this investigation and how to close them as well as indicating the holistic conclusion and recommendations.

    Response: We appreciate your suggestion. Therefore the conclusion paragraph has been changed

Reviewer 4 Report

1.    It is necessary to indicate the yield of raw extracts so that the reader can appreciate the completeness of extraction.

2.    The data in Table 1 are given with different accuracy. It is enough to give one figure after the decimal point, since colorimetric methods are rarely capable of great accuracy.

3.    It is not entirely correct to call the extracts acetone, since the extraction was carried out with aqueous acetone.

4.    The authors should pay special attention to the anticancer activity of polyprenols and triterpenoids, which make a significant contribution to the active properties of plant extracts. For example, in review (Swiezewska et al, 1994), an entire chapter is devoted to the study of polyprenols in plants of the genus Potentilla. Swiezewska, E.; Sasak, W.; MaÅ„kowski, T.; Jankowski, W.; Vogtman, T.; Krajewska, I.; Hertel, J.; Skoczylas, E.; Chojnacki, T. The search for plant polyprenols. Acta Biochim. Pol. 199441, 221–260.

5.    Shrub species of Potentilla are bred into a separate genus Pentaphylloides (Dasiphora), numbering up to 13 species. For phytochemical studies, this is significantly, since the aerial part of shrub species includes bark and wood.

Author Response

Reviewer #4:

  1. It is necessary to indicate the yield of raw extracts so that the reader can appreciate the completeness of extraction.
    Response: Thank you for this valuable suggestion. Therefore, in materials and methods section the Table 4 (section materials and methods) was added with proper information.

  2. The data in Table 1 are given with different accuracy. It is enough to give one figure after the decimal point, since colorimetric methods are rarely capable of great accuracy.
    Response: We appreciate your suggestion. Table 1 has been rearranged.

  3. It is not entirely correct to call the extracts acetone, since the extraction was carried out with aqueous acetone.
    Response: We appreciate your suggestion. The information considering type of solvent used and type of obtained extracts were changed.

  4. The authors should pay special attention to the anticancer activity of polyprenols and triterpenoids, which make a significant contribution to the active properties of plant extracts. For example, in review (Swiezewska et al, 1994), an entire chapter is devoted to the study of polyprenols in plants of the genus Potentilla. Swiezewska, E.; Sasak, W.; MaÅ„kowski, T.; Jankowski, W.; Vogtman, T.; Krajewska, I.; Hertel, J.; Skoczylas, E.; Chojnacki, T. The search for plant polyprenols. Acta Biochim. Pol. 1994, 41, 221–260.
    Response: We appreciate your suggestion. We are aware that polyprenols and triterpenoids isolated from Potentilla species through the last few decades exerted significant anticancer effects, however, Potentilla species were commonly used for the treatment of intestinal disorders and diarrhoea, and the effects are mainly assigned to a rich-polyphenol fraction (underground parts - hydrolysable and condensed tannins and aerial parts - flavonoids). Therefore we conducted a purification process with chloroform (described in 3.2. Plant Materials and Extraction Procedure) to eliminate the lipophilic matrix to uncover the influence of polyphenols simultaneously on normal and cancer colon cells under the same conditions. Moreover, with the employment of LC-HRMS analysis, we found no presence of polyprenols and triterpenoids in tested samples. Therefore we did not include in the discussion the contribution of those compounds regarding the observed effect.

  5. Shrub species of Potentilla are bred into a separate genus Pentaphylloides (Dasiphora), numbering up to 13 species. For phytochemical studies, this is significantly, since the aerial part of shrub species includes bark and wood.
    Response: Thank you for this valuable suggestion. Indeed, Dasiphora species are distinguished from the Potentilla genus, however, the relationship is close and the species for a long time was commonly classified in the Potentilla genus. However, this species is indeed shrub and term “aerial parts” is misleading and therefore proper information considering the names, synonyms and parts used in the study was added in the Table 4 (section materials and methods).

Round 2

Reviewer 1 Report

Thank you

Reviewer 3 Report

The authors satisfactory answered to the reviewer's suggestions. The manuscript is now suitable for publication in IJMS.

Reviewer 4 Report

The authors did not take into account the remark regarding the contribution of polyprenols to anticancer activity. However, this contribution is significant, especially for plant material with a high content of polyprenols (such as Potentilla and other Rosaceae). For example: https://doi.org/10.1016/j.fitote.2015.09.008
